# Contrast-Enhanced Ultrasound (CEUS) in the Evaluation of Renal Masses with Histopathological Validation—Results from a Prospective Single-Center Study

**DOI:** 10.3390/diagnostics12051209

**Published:** 2022-05-12

**Authors:** Antonio Tufano, Francesco Maria Drudi, Flavia Angelini, Eleonora Polito, Milvia Martino, Antonio Granata, Giovanni Battista Di Pierro, Eriselda Kutrolli, Matteo Sampalmieri, Vittorio Canale, Rocco Simone Flammia, Daniele Fresilli, Michele Bertolotto, Costantino Leonardo, Giorgio Franco, Vito Cantisani

**Affiliations:** 1Department of Urology, University La Sapienza, 00185 Roma, Italy; gb.dipierro@libero.it (G.B.D.P.); matteo.sampalmieri@uniroma1.it (M.S.); canale.vittorio@gmail.com (V.C.); roccosimone92@gmail.com (R.S.F.); costantino.leonardo@uniroma1.it (C.L.); giorgio.franco@uniroma1.it (G.F.); 2Department of Radiology, Oncology and Pathology, University La Sapienza, 00185 Roma, Italy; francescom.drudi@uniroma1.it (F.M.D.); flavia.angel91@gmail.com (F.A.); eleonora.polito@uniroma1.it (E.P.); aivlim@hotmail.it (M.M.); eriselda.kutrolli@uniroma1.it (E.K.); daniele.fresilli@uniroma1.it (D.F.); vito.cantisani@uniroma1.it (V.C.); 3Nephrology and Dialysis Unit, “Cannizzaro” Hospital, 95122 Catania, Italy; antonio.granata4@tin.it; 4Department of Radiology, University of Trieste, 34127 Trieste, Italy; bertolot@units.it

**Keywords:** contrast-enhanced ultrasound (CEUS), renal cancer, renal masses, perfusion, quantitative analysis

## Abstract

Background: To evaluate the diagnostic accuracy of contrast-enhanced ultrasound (CEUS) in characterizing between malignant and benign renal lesions confirmed by histological examination. Methods: Overall, 110 patients, for a total of 118 renal masses previously identified at CT and MRI underwent CEUS. An expert radiologist evaluated morphological, qualitative and quantitative parameters. Acquired data were analyzed to assess the value of each parameter to differentiate between malignant and benign lesions. Results: Histological results of 118 renal masses showed 88 (75%) malignant lesions and 30 (25%) benign lesions. Among morphological features, inhomogeneous echogenicity was the best predictor of malignancy depicting a sensitivity, specificity, positive predictive value (PPV) and negative predictive value (NPV) of 76%, 76%, 88% and 57%, respectively. Among qualitative parameters, the most reliable parameter was the presence of pseudo-capsule. Here, sensitivity, specificity, positive PPV and NPV were 85%, 86%, 94% and 71%, respectively. Among quantitative parameters, the most reliable parameters were peak intensity (PI) and the area under the (AUC) with sensitivity, specificity, PPV and NPV values of 94%, 92%, 96% and 87% and 99%, 92%, 97% and 97%, respectively. Finally, the most reliable parameters were combined to distinguish between benign and malignant lesions. The best combination obtained was restricted to CEUS parameters (PI and AUC). Here, sensitivity, specificity, PPV, NPV and accuracy rate were 93%, 100%, 100%, 83% and 93%, respectively. Conclusions: CEUS increases the US accuracy to discriminate between benign and malignant renal lesions.

## 1. Introduction

Renal cell carcinoma (RCC) is the most common malignant tumor involving the genitourinary tract, accounting for 2–3% of all human cancers [1]. Incidence of renal cancer is highest in developed countries with rates generally higher in Europe and North America [2]. Nowadays, most renal tumors encountered in clinical practice are small incidentalomas (cT1a) [3].

Over the last few decades, the widespread use of abdominal imaging has increased the detection of localized, asymptomatic tumors, while less than 10% of patients currently present with the “classic triad” of symptoms: hematuria, flank pain and palpable masses [4]. Nowadays, new diagnostic perspectives are offered by multiparametric ultrasound (US), in which US B-mode is combined with color Doppler investigation (CDI), contrast-enhanced ultrasound (CEUS) and elastography. Specifically, CEUS, allows a dynamic functional imaging of tumor perfusion [5].

The benefit of using microbubble contrast agents in renal disorders has been well documented and the use of CEUS for renal indications has subsequently become part of the European Federation of Societies for Ultrasound in Medicine and Biology (EFSUMB) guidelines in 2008 with the latest update in 2017 [6]. All approved US contrast agents have no renal excretion and renal insufficiency is not a contraindication for their use [7]. Interestingly, CEUS is useful in the diagnosis of kidney lesions and during the follow-up of non-surgical lesions, representing a cost-effective and non-invasive method, not requiring the use of ionizing radiation.

Moreover, previous studies showed that CEUS can be used to differentiate among lesions with an equivocal enhancement at CT or MRI [8,9,10,11].

Therefore, CEUS has a potential role in the evaluation of renal lesions that deserve further investigation. Aim of this study is to evaluate the diagnostic usefulness of qualitative and quantitative CEUS for renal masses characterization as confirmed by histological examination.

## 2. Materials and Methods

### 2.1. Study Design, Inclusion and Exclusion Criteria

Between December 2019 and September 2021, a prospective single-center study was conducted at our institution. A total of 110 consecutive patients were enrolled. Overall, 118 renal masses, previously confirmed with CT or MRI, were studied with CEUS. Informed consent for the contrast-enhanced study was obtained from each patient. The study protocol was approved by the local ethical committee and was in accordance with the Helsinki Declaration of 1975. Inclusion criteria were: (1) renal mass detected by standard US; and (2) histopathological confirmation used as reference standard. Exclusion criteria were: (1) patients with contraindication for a contrast-enhanced study; (2) patients diagnosed with simple cysts on US B-mode; and (3) patients with metastases detected on preoperative imaging.

### 2.2. Imaging Acquisition and Interpretation

All baseline US and contrast enhancement ultrasound (CEUS) studies have been performed, with the same high-end US equipment (Toshiba Alpio 800, Tokyo, Japan) using 3.5-5-5 MHz convex probe. CEUS examination was performed by an expert radiologist with more than 20 years’ experience in genito-urinary ultrasound imaging. A 1 mL microbubble second generation contrast agent (SonoVue, Bracco Imaging, Milan, Italy) was administered via intravenous injection, followed by 5–10 mL of saline solution 0.9%. Microbubble enhancement has been followed continuously for at least 2 min. According to EFSUMB guidelines, a real-time video clip was recorded for each patient for review and documentation. After the completion of the imaging investigation, patients underwent surgical procedure of nephron sparing surgery (NSS). Histopathological examination has been performed and used asa reference standard in this study.

### 2.3. Morphological Pattern Analysis with US B-Mode and Color Doppler

Preliminary conventional US and CDI were performed, describing the lesion site, dimension, margin, echo-pattern and vascularization. According to US patterns, we classified renal masses presenting with regular margins and homogeneous hyperechoic or hypoechoic echogenicity as potentially benign. Masses presenting with irregular margins, inhomogeneous hyperechoic or hypoechoic echogenicity and/or with mixed appearance were considered potentially malignant. CDI patterns were classified as follows: type 1 (absence of intra-lesion vascularization); type 2 (peripheral vascularization); and type 3 (peripheral and intra-lesion vascularization). Type 1 has been considered benign, type 2 and 3 have been considered potentially malignant.

### 2.4. Qualitative CEUS Analysis

Parameters evaluated were: (1) The enhancement of the lesion, classified as rapid wash-in or with synchronous/late wash-in compared to surrounding healthy renal parenchyma; (2) homogeneous or inhomogeneous wash-in; and (3) the presence or absence of pseudo-capsule, defined as rim-enhancement. Potentially malignant parameters considered were the presence of rapid and inhomogeneous wash-in and rapid wash-out, when compared to healthy renal parenchyma. Potentially benign parameters considered were wash-in homogeneous and synchronous/late, when compared to healthy renal parenchyma.

### 2.5. Quantitative CEUS Analysis

Prospectively, videos recorded during CEUS have been used for elaborating time/intensity curves (TIC). Regions of interest (ROI) have been placed inside the lesion and within healthy renal tissue. Quantitative parameters of TIC have been automatically calculated by the software integrated in the US equipment. Finally, four different parameters were computed: (1) Peak of the signal intensity (PI) defined as maximum intensity value in time-intensity curve; (2) time to peak (sec) (TP) defined as the time elapsed between the moment when contrast medium first reaches the lesion and the time of maximum signal intensity after contrast medium administration; (3) mean transit time (sec) (MTT), defined as the lesion’s enhancement duration; and (4) AUC (area under the curve).

### 2.6. Statistical Analysis

Continuous variables were expressed as median and interquartile range (IQ), and discrete variables as numbers and percentages. Statistical analyses were performed using Stata 15.0 calculating sensibility, specificity, positive predictive value (PPV), negative predictive value (NPV) and area under the curve (AUC) of every parameter with dichotomous outcome. Reliability of measured parameters on a continuous scale has been calculated using the analysis of the ROC curve, in which the AUC has been measured and the Youden test has been performed to find out the cut-off to maximize diagnostic accuracy. The match between curves has been accomplished using the Bonferroni test with a statistical significance level of less than 0.05.

## 3. Results

### 3.1. Population Characteristics

Overall, 110 patients (8 presenting with multiple masses) with a total of 118 lesions were evaluated by US B-mode, CDI and CEUS. Of those, 88 (75%) were malignant and 30 (25%) were benign. Patients and tumor characteristics are described in Table 1. All patients underwent NSS. Median age was 63 years (IQ: 55–71) and the median tumor size was 2.85 cm (range: 1.5–3.9 cm) with a median R.E.N.A.L. score of 7 (IQ: 6–9). Histologically, 88 lesions were classified as malignant (72 clear cell renal cell carcinoma (Figure 1), 6 type I papillary renal cell carcinoma, 4 clear cell cystic carcinoma, 4 clear cell multicystic carcinoma and 2 chromophobe renal cell carcinoma) and 30 as benign lesions (16 oncocytoma, 10 angiomyolipoma, 2 renal abscess and 2 hemorrhagic cyst) (Table 2).

### 3.2. Diagnostic Value of Ultrasound Parameters

Regarding US and CDI evaluation (Table 3), the most reliable parameter suspicious for malignancy was inhomogeneous echogenicity. Here, sensibility, specificity, PPV and NPV were 76%, 76%, 89% and 58%, respectively. Accuracy rate was 77%. Subsequently, peripheral and intra-lesional vascularization showed a sensibility, specificity, PPV and NPV of 70%, 57%, 79% and 45%, respectively. Accuracy rate was 64%. Notably, irregular margins depicted a poor accuracy of 46% so they were not included as a potential marker of malignancy.

Using the Bonferroni test and taking inhomogeneous echogenicity as a reference, a statistically significant difference was found in comparison with irregular margins (*p* < 0.05).

### 3.3. Diagnostic Values of Qualitative CEUS Parameters

Among the qualitative CEUS parameters, the most reliable pattern of malignant lesion was the presence of pseudo-capsule. Here, sensibility, specificity, PPV and NPV were 85%, 86%, 94% and 71%, respectively. Diagnostic accuracy rate was 86%. Moreover, rapid and inhomogeneous wash-in depicted an accuracy of 76% and 77%, respectively. Here, sensibility, specificity, PPV and NPV were 82%, 70%, 87%, 62% and 76%, 78%, 89% and 58%, respectively. Notably, rapid wash-out depicted a poor accuracy (67%) and was not included as a potential marker of malignancy (Table 4).

The pseudo-capsule showed a statistical significance difference in comparison with rapid wash-out (*p* < 0.05).

### 3.4. Diagnostic Values of Quantitative CEUS Parameters

Among the quantitative CEUS parameters, the most reliable patterns of malignant lesion were PI and AUC. Here, sensitivity, specificity, PPV and NPV were 94%, 92%, 96%, 87% and 99%, 92%, 97%, 97%, respectively. Diagnostic accuracy rates were 93% and 95%, respectively (Table 5). Using the Bonferroni test and taking AUC as a reference, statistically significant differences were found in comparison with TP and MTT (*p* < 0.001).

### 3.5. Diagnostic Values of the Combined Parameters (US + CEUS)

Finally, a combination of all the best reliable patterns (echogenicity, PI and AUC) depicted a sensitivity, specificity, PPV, NPV and accuracy of 73%, 100%, 100%, 56% and 81%, respectively, in the identification of malignant lesions. Moreover, when the analysis was restricted to CEUS parameters (PI and AUC), the diagnostic performance improved, achieving a sensitivity, specificity, PPV, NPV and accuracy of 93%, 100%, 100%, 83% and 93%, respectively (Table 6).

## 4. Discussion

CEUS represents a promising tool in the characterization of renal masses. In this study, the potential of this technique in a consecutive series of patients with renal masses has been prospectively investigated.

Our analyses resulted in several noteworthy observations.

First, we identified 110 patients with a diagnosis of renal masses treated by NSS. Of those, 8 patients presented with multiple masses, in consequence a total of 118 lesions were analyzed. Overall, the median tumor size was 2.8 cm and median R.E.N.A.L. nephrometry score was 7. Tumor lesions were sided either right (*n* = 58) or left (*n* = 60) and located at the superior (*n* = 27), middle (*n* = 40) and inferior (*n* = 51) third of the kidney. Overall, renal masses were more frequently malignant vs. benign 75% vs. 25%, respectively. These histopathological findings are in line with other published series [12].

Second, we analyzed the diagnostic performance of US B-mode parameters (echogenicity, vascularization and tumor margin) to distinguish malignant vs. benign renal masses. Here, inhomogeneous echogenicity resulted in the best accuracy (76%) relative to either vascularization (64%) or tumor margin (46%). Moreover, echogenicity characteristics resulted in interestingly high PPV (88%), but low NPV (57%). Notably, differentiating between US, benign solid renal masses from RCC remains a challenge and echogenicity of these masses is heterogeneously described in the literature [13].

Third, we analyzed the diagnostic performance of CEUS qualitative parameters (presence of pseudo-capsule, inhomogeneous wash-in, rapid wash-in and rapid wash-out) in distinguishing malignant vs. benign renal masses. Here, the presence of pseudo-capsule resulted in the best accuracy (86%) relative to either rapid wash-in (76%) or inhomogeneous wash-in (77%) or rapid wash-out (67%). Moreover, the presence of pseudo-capsule resulted in interestingly high PPV and NPV of 94% and 71%, respectively. These observations are thus in agreement with the findings of previous studies, where the presence of a perilesional rim of enhancement resulted useful especially when distinguishing RCC from angiomyolipomas [14,15]. Moreover, homogeneous hypo- or iso-enhancement and rapid wash-in were associated with benign features in other historical studies [16,17].

Fourth, we analyzed the diagnostic performance of CEUS quantitative parameters, such as PI, AUC, MTT and TP. Our analyses did not allow us to set an adequate cut-off for both MTT and TP, and for this reason they were not included. Conversely, both PI and AUC yielded an accuracy of 93% and 95%, respectively. Moreover, AUC resulted in interestingly high PPV and NPV of 97% and 97%, respectively. On the other hand, PI showed a similar PPV of 96%, but a lower NPV of 87%. As a consequence, higher PI and increased AUC were two key differentiating features in predicting malignancy in the present study. These results were previously confirmed by Dai et al., where PI values were significantly higher in malignant lesions [18]. By contrast, Xue et al. reported that TP outperformed PI in detecting malignancy [19].

Fifth, we combined parameters with the highest accuracy in US B-mode (echogenicity) and in CEUS (PI and AUC). This combined score yielded a PPV of 100 % and NPV of 56% with an accuracy of 81%. Finally, after removing echogenicity, the combination of PI and AUC yielded a PPV of 100% and NPV of 83% with an accuracy of 93%. Taken together, CEUS parameters showed high values of diagnostic accuracy. Conversely, B-mode US parameters yielded a low accuracy rate with the exception of echogenicity. In consequence, we recorded an additional benefit when adding CEUS parameters in the analyses of renal masses. These results are promising, since both PI and AUC are quantitative parameters and for this reason are widely reproducible. Here, we adopted cut-offs calculated in our preliminary analyses (Youden test). Nevertheless, these cut-offs need to be validated in an external cohort to test their reliability.

Our work in not devoid of limitations and should be interpreted in the context of its single-center cohort. First, the small sample size and relatively small percentage of benign tumors, only 25% of our sample. Second, we only enrolled patients with an indication for surgery, in consequence, we potentially excluded all those patients undergoing active surveillance from the study. Third, all cases were evaluated by a single reader with long experience in urological ultrasound imaging. For this reason, our results may overestimate the diagnostic accuracy of this technique and potentially undermine their reproducibility in clinical practice. Further studies are needed to evaluate the diagnostic accuracy of CEUS in characterizing renal masses and to assess inter-reader variability for sonologists with different expertise.

## 5. Conclusions

Our findings suggest that CEUS is a promising additional diagnostic tool capable of distinguishing between malignant vs. benign renal masses. The high degree of diagnostic accuracy depicted can play a role in the clinical decision-making strategy of localized renal masses.

## Figures and Tables

**Figure 1 diagnostics-12-01209-f001:**
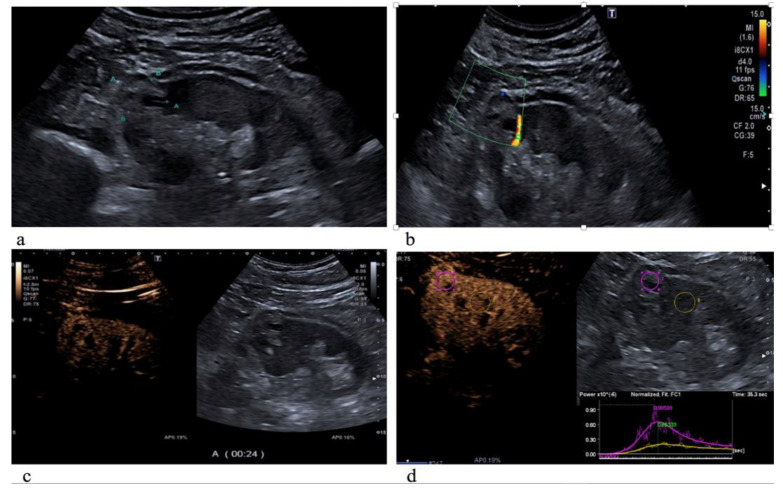
Clear cell carcinoma on the upper pole of the left kidney, characterized by an inhomogeneous hyperechoic echotexture with fluid-filled lacuna (**a**); on color Doppler US evaluation, it appeared with poor peripheral vascularization (**b**); and on CEUS evaluation, the lesion showed rapid and inhomogeneous enhancement compared to the normal renal cortex parenchyma (**c**). As showed by the time-intensity curve (**d**), the renal mass has a higher peak of the signal intensity (PI) and an incremented regional blood volume (RBV) in comparison to the surrounding healthy parenchyma.

**Table 1 diagnostics-12-01209-t001:** Baseline data.

Characteristics		Overall	Benign	Malignant
		*n* = 118	*n* = 30 (25%)	*n* = 88 (75%)
**Age at diagnosis, yrs**	Median (IQ)	63 (51–71)	63 (53–71)	64 (49–71)
**Gender, male**	*n* (%)	73 (61.8)	19 (63.1)	50 (56.8)
**Laterality, right**	*n* (%)	60 (50.8)	18 (60.0)	42 (47.7)
**Tumor location**	*n* (%)			
Upper pole		27 (22.9)	9 (30)	18 (20.4)
Middle part		40 (33.9)	7 (23.3)	33 (37.6)
Lower pole		51 (43.2)	14 (47.7)	37 (42.0)
**Tumor size, cm**	Median (IQ)	2.8 (1.5–3.9)	3.1 (2.2–3.7)	2.6 (1.4–3.9)
**R.E.N.A.L score**	Median (IQ)	7 (6–9)	7 (5–8)	8 (6–9)

**Table 2 diagnostics-12-01209-t002:** Histopathological characteristics.

Benign	*n* (%)	Malignant	*n* (%)
Oncocytoma	16 (53.3)	Clear cell	72 (81.8)
Angiomylipoma	10 (33.3)	Papillary	6 (6.9)
Kidney abscess	2 (6.7)	Cystic clear cell	4 (4.5)
Hemorrhagic cyst	2 (6.7)	Multicystic clear cell	4 (4.5)
		Chromophobe	*2 (2.3)*

**Table 3 diagnostics-12-01209-t003:** Diagnostic values of ultrasound parameters (SEN: Sensitivity; SPE: Specificity; ROC: Receiver Operating Characteristics; PPV: Positive Predictive Value; NPV: Negative Predictive Value).

	SEN % (CI)	SPE % (CI)	ROC % Area	PPV % (CI)	NPV % (CI)
Margins [irregular]	**29** (20–40)	**62** (45–77)	**46** (37–55)	**65** (48–79)	**27** (18–38)
Echogenicity [inhomogeneous]	**76** (66–85)	**76** (59–88)	**76** (68–84)	**88** (79–94)	**57** (42–71)
Vascularization [type 2 or 3]	**70** (59–80)	**57** (39–73)	**64** (54–73)	**79** (69–88)	**45** (30–60)

**Table 4 diagnostics-12-01209-t004:** Diagnostic performance of qualitative evaluation performed by CEUS (SEN: Sensitivity; SPE: Specificity; ROC: Receiver Operating Characteristics; PPV: Positive Predictive Value; NPV: Negative Predictive Value).

	SEN % (CI)	SPE % (CI)	ROC % Area	PPV % (CI)	NPV % (CI)
Wash-in [rapid]	**82** (72–89)	**70** (50–84)	**76** (68–84)	**87** (77–93)	**62** (46–76)
Wash-in [inhomogeneous]	**76** (66–85)	**78** (62–90)	**77** (69–85)	**89** (80–95)	**58** (43–72)
Wash-out [rapid]	**53** (42–64)	**81** (65–92)	**67** (59–75)	**87** (75–95)	**42** (31–55)
Pseudo-capsule	**85** (76–92)	**86** (71–95)	**86** (79–92)	**94** (86–98)	**71** (56–84)

**Table 5 diagnostics-12-01209-t005:** Diagnostic performance of quantitative evaluation performed by CEUS (SEN: Sensitivity; SPE: Specificity; ROC: Receiver Operating Characteristics; PPV: Positive Predictive Value; NPV: Negative Predictive Value; PI: Peak Intensity; AUC: Area Under the Curve).

	SEN % (CI)	SPE % (CI)	ROC % Area	PPV % (CI)	NPV % (CI)
PI	**94** (87–98)	**92** (78–98)	**93** (88–98)	**96** (90–99)	**87** (72–96)
AUC	**99** (94–100)	**92** (78–98)	**95** (91–100)	**97** (91–99)	**97** (85–100)

**Table 6 diagnostics-12-01209-t006:** Diagnostic performance of the combined best parameters (SEN: Sensitivity; SPE: Specificity; ROC: Receiver Operating Characteristics; PPV: Positive Predictive Value; NPV: Negative Predictive Value; PI: Peak Intensity; AUC: Area Under the Curve).

	**SEN % (CI)**	**SPE % (CI)**	**ROC % Area**	**PPV % (CI)**	**NPV % (CI)**
Echogenicity [inhomogeneous] + PI + AUC	**73** (68–78)	**100** (95–100)	**81** (74–85)	**100** (95–100)	**56** (43–70)
PI + AUC	**93** (88–98)	**100** (95–100)	**93** (88–98)	**100** (95–100)	**83** (77–86)

## Data Availability

The data presented in this study are available from the corresponding author upon request.

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
