# Peer review of "Contrast-Enhanced Ultrasound (CEUS) in the Evaluation of Renal Masses with Histopathological Validation—Results from a Prospective Single-Center Study"

_diagnostics, 2022, doi:10.3390/diagnostics12051209_

Round 1

Reviewer 1 Report

This paper presents a study of sensitivity and specificity of ultrasound imaging of suspicious renal masses.
The question is to find the optimal parameters of the contrast enhanced ultrasound with injections of microbubble contrast agent which have already shown their interest in this context.
The experimental and statistical methodologies are simple but effective and makes it possible to determine the most discriminating elements of the CEUS in this particular context of for the population studied.
As stated by the authors, a study on another population will be necessary to validate the performance of these diagnostic criteria.
The manuscript does not specify if certain variables (contrast homogeneity, or else) may help to discriminate between the variuous malgnant histological types.

Author Response

Dear Reviewer,

Thank you for your constructive comments on our manuscript.

Below you find our answers:

1)     As stated by the authors, a study on another population will be necessary to validate the performance of these diagnostic criteria.

We totally agree with this observation. We are actually enrolling new patients to validate in the next future our results and perform new analysis.

      2) The manuscript does not specify if certain variables (contrast homogeneity, or else) may help to discriminate between the various malignant histological types.

Thank you for this comment. In our analysis of qualitative parameters, wash-in (inhomogeneous or rapid) was one of the parameters used for malignancy. In paragraph 2.4 (materials and method) we corrected and modified the word enhancement with wash-in.

We hope that paragraph 2.4 is exhaustive for this point since we evaluated both wash-in and wash-out of the lesions.

Reviewer 2 Report

The authors conducted a prospective study to evaluate the role of contrast-enhanced ultrasound (CEUS) in the diagnosis of renal malignancy A total of 118 lesions with pathologically confirmed masses were assessed. Three evaluation systems (classical US combined with color Doppler investigation; quantitative CEUS and qualitative CEUS) were employed and analyzed. They concluded that CESU measurement for renal mass had the additional benefits than traditional US. Several concerns are listed as below:

  1. Totally there were 118 lesions in 110 patients. What is the details? How about patients with bilateral lesions or unilateral lesion?
  2. What is R.E.N.A.L score? Does it play any role in predicting malignancy in this study?
  3. Whether there was distant metastasis was not mentioned in this study.
  4. As most of the malignancy was clear cell, did the authors analyze whether different types malignancy had different predicting pattern? For example, clear cell vs. papillary type.
  5. Similarly, whether tumor size is another factor that could affect the predicting ability and accuracy.
  6. In the combination analysis, the authors found that accuracy rate in US-B-mode combined with CEUS was 81%, but is was 93% when using PI and AUC only. The authors should discuss on this finding to explain how to interpret abnormal findings in traditional US and CEUS respectively.
